# Nuclear Binding Protein 2/Nesfatin-1 Affects Trophoblast Cell Fusion during Placental Development via the EGFR-PLCG1-CAMK4 Pathway

**DOI:** 10.3390/ijms25031925

**Published:** 2024-02-05

**Authors:** Qinyu Dang, Yandi Zhu, Yadi Zhang, Zhuo Hu, Yuchen Wei, Zhaoyang Chen, Xinyin Jiang, Xiaxia Cai, Huanling Yu

**Affiliations:** 1Department of Nutrition and Food Hygiene, School of Public Health, Capital Medical University, Beijing 100069, China; dang_qinyu@mail.ccmu.edu.cn (Q.D.); zhuyandi@ccmu.edu.cn (Y.Z.); yadizhang@mail.ccmu.edu.cn (Y.Z.); huzhuo99@mail.ccmu.edu.cn (Z.H.); weiyuchen_98@mail.ccmu.edu.cn (Y.W.); chenzhaoyang@mail.ccmu.edu.cn (Z.C.); caixx1988@ccmu.edu.cn (X.C.); 2Departments of Health and Nutrition Sciences, Brooklyn College of City University of New York, New York, NY 11210, USA; xinyinjiang@brooklyn.cuny.edu

**Keywords:** trophoblast fusion, placenta, *NUCB2*/Nesfatin-1, EGFR/PLCG1/CAMK4, embryonic development

## Abstract

Previous studies have shown that nuclear binding protein 2 (*NUCB2*) is expressed in the human placenta and increases with an increase in the syncytialization of trophoblast cells. This study aimed to investigate the role of *NUCB2* in the differentiation and fusion of trophectoderm cells. In this study, the expression levels of *NUCB2* and E-cadherin in the placentas of rats at different gestation stages were investigated. The results showed that there was an opposite trend between the expression of placental *NUCB2* and E-cadherin in *rat* placentas in different trimesters. When primary human trophoblast (PHT) and BeWo cells were treated with high concentrations of Nesfatin-1, the trophoblast cell syncytialization was significantly inhibited. The effects of *NUCB2* knockdown in BeWo cells and Forskolin-induced syncytialization were investigated. These cells showed a significantly decreased cell fusion rate. The mechanism underlying *NUCB2*-regulated trophoblast cell syncytialization was explored using RNA-Seq and the results indicated that the epidermal growth factor receptor (*EGFR*)-phospholipase C gamma 1 (*PLCG1*)-calmodulin-dependent protein kinase IV (*CAMK4*) pathway might be involved. The results suggested that the placental expression of *NUCB2* plays an important role in the fusion of trophoblasts during differentiation via the *EGFR-PLCG1-CAMK4* pathway.

## 1. Introduction

During human pregnancy, normal differentiation, proliferation, migration, and infiltration of trophoblast cells are essential for embryo implantation and placenta formation [1,2]. In particular, the syncytialization of trophoblast cells plays a critical role in the embryonic and placental development. Premature birth, fetal growth limitation, and pregnancy complications have all been linked to aberrant syncytialization and syncytiotrophoblast function [3,4]. To create a multinucleated syncytiotrophoblast (STB), mononucleated cytotrophoblast (CTB) cells must unite. The STB layer of the human placenta is a syncytial structure with endocrine, immune, and nutritional functions which plays an important role in the exchange of nutrients and gases between maternal and fetal blood circulation [5]. Two major cellular models are used to study trophoblast cell syncytialization in vitro. One model uses the primary human trophoblast cells that are directly isolated from the human placenta and spontaneously syncretizes, and the other is a syncytialization model of Forskolin-induced BeWo choriocarcinoma cells (Forskolin–BeWo cells) [3]. Human chorionic gonadotropin (hCG), human placental prolactin, glial cell missing transcription factor 1 (*GCM1*), syncytin-1, and syncytin-2 have been reported to be associated with trophoblast syncytialization [6,7]. These syncytial markers’ abnormal expression has been linked to poor syncytialization and impairs placental growth and function, leading to fetal growth restriction and pregnancy problems [8]. However, the molecular mechanisms of trophoblast syncytialization still need to be explored.

Nesfatin-1 is a macro-nutrient metabolism regulatory peptide produced from the N-terminal fragment and active center of hydrolytic nuclear binding protein 2 (*NUCB2*), consisting of 82 amino acids [9,10]. *NUCB2*/Nesfatin-1 is expressed in the hypothalamus, liver, adipose tissue, uterus, and placenta, with different abundances in different tissues [11,12,13]. Numerous studies have indicated the critical role of *NUCB2*/Nesfatin-1 in various metabolic functions, including lipid and glucose metabolism, reproduction, and gastrointestinal function [14]. In previous studies, it was found that maternal serum Nesfatin-1 levels were significantly lower in pre-eclampsia compared to normal pregnancy and that pre-eclampsia (PE) severity and presence were inversely correlated with serum Nesfatin-1 levels [15,16]. The expression of placental *NUCB2* was significantly reduced in late pregnancy during rapid fetal weight gain [17]. Nesfatin-1 was found to inhibit oxidative stress through the activation of the PI3K/AKT/mTOR and AKT/GSK3β signaling pathways, thereby promoting the proliferation, migration, and invasion of trophoblast cells [18]. *NUCB2* was found to be expressed mainly in STBs and increased during syncytialization of primary human placental trophoblast cells [14]. It was also found that *NUCB2* expression increased significantly after differentiation in a Forskolin-induced BeWo differentiation fusion model [19]. Therefore, *NUCB2* may be involved in the syncytialization of trophoblast cells. *NUCB2* was a positive regulator of EGF-dependent signaling, leading to enhanced cell growth and inhibition of adipocyte differentiation [20]. It was also found that epidermal growth factor receptor is highly abundant in the placenta and plays an important role in cell proliferation, differentiation, invasion, and fusion [21,22]. When epidermal growth factor receptor (*EGFR)* activation is compromised, it further affects trophoblast cell fusion [23]. A study demonstrated that calmodulin-dependent protein kinase IV (*CAMK4)* is closely related to trophoblast function. It was also found that *EGFR* affects the activation of phospholipase C gamma 1 (*PLCG1*) and *CAMK4* [24]. However, the role and mechanism of *NUCB2*/Nesfatin-1 in human placental syncytial disorder requires further study.

To further understand if and how *NUCB2*/Nesfatin-1 regulates trophoblast differentiation, changes in *NUCB2* in *rat* placentas at different stages of pregnancy were investigated. Then, *NUCB2* expression levels in Forskolin–BeWo cells and in a primary human trophoblast spontaneous syncytial model were investigated. Next, the regulatory effects of exogenous Nesfatin-1 on the differentiation and fusion of primary human trophoblast and Forskolin–BeWo cells were explored. Changes in differentiation and fusion were investigated using a BeWo cell model with a stable knockdown of *NUCB2*, and the preliminary mechanisms by which *NUCB2* affects trophoblast differentiation and fusion were explored.

## 2. Results

### 2.1. NUCB2 Expression at Different Stages of Gestation in the Rat Placenta

The expression of E-cadherin, a key adhesion molecule associated with syncytialization, was significantly reduced in the trophoblasts of fused cells. Using double immunofluorescence labeling, we observed consistent changes in the fluorescence area of E-cadherin and *NUCB2* in the labyrinthine and connective areas from gestational days 14.5 to 19.5. There was a significant decrease on gestational day 17.5 compared to that on gestational day 14.5 (*p* < 0.001), and a significant increase on gestational day 19.5 compared to that on gestational day 17.5 (*p* < 0.001). Meanwhile, RT-PCR and Western blot were used to detect the expression of E-cadherin mRNA and protein in *rat* placentas at different gestation stages. E-cadherin mRNA expression fluctuated between gestation days 14.5 and 19.5, peaking on day 19.5 (Figure 1b). Meanwhile, the level of E-cadherin protein similarly showed a trend of steady increase, peaking on day 19.5 (Figure 1c).

In contrast, the fluorescence area of *NUCB2* increased from gestational day 14.5 to gestational day 19.5 in both the labyrinthine and junctional regions (Figure 1a). Meanwhile, RT-PCR and Western blot detected the expression of *NUCB2* mRNA and protein in *rat* placentas at different times. Placental *NUCB2* mRNA levels showed a clear trend of increasing and then decreasing from gestation days 14.5 to 19.5, with *NUCB2* mRNA levels on gestation day 17.5 being significantly higher than that those on gestation days 14.5 and 19.5, and reaching the lowest level on gestation day 19.5 (Figure 1d). In contrast, the placental *NUCB2* protein levels decreased continuously from gestational days 14.5 to 19.5, reaching the lowest level on gestational day 19.5. The protein levels of *NUCB2* on gestational day 19.5 were significantly lower (*p* < 0.05) compared to both gestational day 14.5 and gestational day 17.5 (Figure 1e).

### 2.2. NUCB2 Expression during Spontaneous Syncytialization of Primary Human Trophoblasts and Forskolin-Induced BeWo Syncytialization

Primary human trophoblast cells were isolated from healthy term placentas and cultured for 72 h. The success of the spontaneous syncytialization model was confirmed by calculating the fusion rate every 24 h using immunofluorescence staining with E-cadherin and by measuring the mRNA and protein levels of the syncytial marker HCG-β every 24 h. The calculation of the fusion rate by means of immunofluorescence revealed an increase in the level of primary trophoblast cell fusion from 24 h onwards. Meanwhile, HCG-β mRNA was assessed using real-time RT-PCR, and HCG-β protein was measured using Western blot. The mRNA and protein levels gradually increased over time (*p* < 0.05) (Appendix A). Successful spontaneous syncytialization of primary human trophoblast cells was confirmed, which was in agreement with previous reports. The model of Forskolin-induced syncytialization of BeWo cells was successfully built, as shown by the immunofluorescence staining with E-cadherin used to calculate the fusion rate of Forskolin-induced BeWo cells per 24 h and the mRNA and protein levels of HCG-β, a syncytialization marker, per 24 h of Forskolin induction. Western blot and RT-PCR were used to detect HCG-β. As the Forskolin induction period increased, the mRNA and protein levels of HCG-β significantly increased over time (*p* < 0.05) (Appendix A).

Through immunofluorescence staining of *NUCB2* at different time points in primary trophoblast cells, it was observed that the average fluorescence intensity of *NUCB2* continuously increased with the syncytialization process of primary trophoblast cells (Figure 2a). RT-PCR and Western blot analysis were used to identify *NUCB2* mRNA and protein levels in these cells at 0 h, 24 h, 48 h, and 72 h in vitro. An increase in *NUCB2* mRNA expression over time was observed (*p* < 0.01) (Figure 2c). *NUCB2* protein levels in cell lysates also significantly increased from 24 h onwards (Figure 2d). These results suggest that *NUCB2* increased during spontaneous syncytialization of primary trophoblast cells.

Immunofluorescence staining of *NUCB2* in Forskolin-induced BeWo cells at different time points was performed, and it was observed that the average fluorescence intensity of *NUCB2* increased consistently during the intervention time (Figure 2b). *NUCB2* mRNA and protein levels in Forskolin-induced BeWo cells were both increased over time (*p* < 0.01) (Figure 2e, and Figure 2f, respectively).

### 2.3. Effect of Exogenous Nesfatin-1 on Forskolin-Induced Fusion of BeWo Syncytized Trophoblast Cells

To investigate the impact of exogenous Nesfatin-1 on the differentiation of trophoblast cells, Nesfatin-1 at concentrations of 4 ng/mL, 20 ng/mL, 100 ng/mL, and 500 ng/mL was used to treat BeWo cells for 4 h. The fusion rate of Forskolin-treated BeWo cells was measured by means of E-cadherin immunostaining. The results revealed that treatment with higher concentrations of Nesfatin-1 (20, 100, and 500 ng/mL) significantly reduced the fusion rate of BeWo cells (*p* < 0.05) (Figure 3a). After the treatment with Nesfatin-1, the placental alkaline phosphatase (PLAP) activity of BeWo cells was significantly lower in the 100 and 500 ng/mL groups compared to the control group (*p* < 0.01) (Figure 3c). The expression levels of ERVW-1, ERVFRD, GCM1, and HCG-β proteins were examined. The protein levels of GCM1 and HCG-β were reduced in the 100 ng/mL and 500 ng/mL Nesfatin-1 groups compared to the control group. The protein levels of ERVW-1 and ERVFRD decreased significantly in all Nesfatin-1-treated groups compared to the control group (*p* < 0.05) (Figure 3b).

### 2.4. Effect of Exogenous Nesfatin-1 on Spontaneous Syncytial Processes in Primary Human Trophoblast Cells

The effects of different concentrations of Nesfatin-1 (4 ng/mL, 20 ng/mL, 100 ng/mL, and 500 ng/mL) on the fusion of human primary trophoblast cells were investigated. Similarly, E-cadherin immunostaining was used to determine the fusion rate of primary trophoblast cells. It was found that treatments with higher concentrations of Nesfatin-1 (20, 100, and 500 ng/mL) significantly reduced the fusion rate of primary trophoblast cells (*p* < 0.05) compared to the control (Figure 4a). The PLAP activity of primary trophoblast cells was significantly lower (*p* < 0.05) in the 500 ng/mL group only when compared to the control group (Figure 4c). The protein levels of GCM1 and HCG-β were reduced in the 100 ng/mL and 500 ng/mL Nesfatin-1 groups compared to the control group. ERVW-1 protein expression was also found to be lower in the 500 ng/mL group (*p* < 0.05) compared to the control group (Figure 4b).

### 2.5. Effect of Knockdown of NUCB2 on the Extent of Trophoblast Fusion in BeWo Cells

Since *NUCB2* was significantly increased during trophoblast syncytialization, it was further investigated whether *NUCB2* expression was essential for the syncytialization phenomenon. A *NUCB2* stable knockdown BeWo cell model was successfully established via lentiviral vector infection. The results showed that *NUCB2* was successfully knocked down at both the RNA and protein levels, as measured by means of immunofluorescence staining, RT-PCR, and Western blotting, respectively (Figure 5a–c). Normal cells, siNC cells (control group), and si*NUCB2* (*NUCB2* knockdown group)-transduced BeWo cells were treated with Forskolin for 48 h for follow-up studies. 

To verify whether *NUCB2* knockdown affected the differentiation and fusion of BeWo cells, changes in a series of metrics related to differentiation and fusion were measured. The calculation of the fusion rate of trophoblast cells was performed using human trophoblast cells stained with anti-E-cadherin and DAPI. According to the immunofluorescence analysis, the siNUCB2 group’s fusion rate was significantly lower than that of the siNC control group and normal cells (*p* < 0.05, Figure 5e). The relative gene expression of *ERVW-1, ERVFRD, GCM1*, and HCG-β was also assessed. The results showed that the mRNA levels of HCG-β and *ERVW-1* were significantly lower in the *siNUCB2* group compared to the siNC group (*p* < 0.05) (Figure 5f). The Western blot results showed that the protein levels of *ERVFRD* and HCG-β were significantly lower in the siNUCB2 group compared to the siNC group (*p* < 0.05) (Figure 5g). The protein levels of ERVW-1, HCG-β, and GCM1 were significantly lower in the siNUCB2 group compared to the normal cell group (*p* < 0.05) (Figure 5g). The PLAP activity was significantly lower in the siNUCB2 group compared to both the normal cell group and the siNC group (*p* < 0.05) (Figure 5d).

### 2.6. NUCB2/Nesfatin-1 May Inhibit Trophoblast Differentiation and Fusion through the EGFR-PLCG1-CAMK4 Pathway

To determine the overall changes in gene expression induced by *NUCB2* knockdown during the differentiation and fusion of BeWo cells, RNA-seq analysis was performed on lentiviral control and lentiviral knockdown groups after cultivation in a 50 μM Forskolin culture for 48 h. The RNA-seq results showed that there were 619 differentially expressed genes between the lentiviral control and the *NUCB2* knockdown groups, of which 240 were upregulated and 279 were downregulated (Figure 6a). *CGB3, CGB5, ERVFRD, and ERVW-1* were significantly down-regulated in the *NUCB2* knockdown group, according to RNA-Seq analysis (Figure 6a), and the GO function enrichment results revealed that the cell differentiation function was significantly different between the two groups (Figure 6b), indicating that *NUCB2* was closely related to the trophoblast differentiation function. The calcium signaling pathway, on the other hand, was downregulated in the si*NUCB2* group, according to KEGG signaling pathway enrichment results (Figure 6c). Epidermal growth factor receptor was discovered to be closely associated with genes such as differentiation and fusion genes (*ERVW-1, CGB, and ERVFRD*) when we clustered the protein–protein interactions pertaining to genes involved in the calcium signaling pathway genes in the STRING database (Appendix A). *EGFR* has been identified as a major component of this signaling network. 

The RT-PCR analysis further validated the EGFR-related pathway genes, and the *EGFR-PLCG1-CAMK4* change trend was found to be consistent with the RNA-seq results. Specifically, the mRNA levels of EGFR, PLCG1, and CAMK4 mRNA were significantly lower in the siNUCB2 group compared to the siNC group (*p* < 0.05, Figure 7a). Similarly, Western blot analysis revealed that the total EGFR and total PLCG1 protein levels in the siNUCB2 group were significantly lower than those in the siNC group (*p* < 0.05), whereas both the total amount and the phosphorylation levels of *CAMK4* in the siNUCB2 group were significantly lower than those in the control group (*p* < 0.05, Figure 7b,c). Therefore, we hypothesized that *NUCB2* might regulate the fusion of trophoblast cell differentiation through the *EGFR-PLCG1-CAMK4* signaling pathway (Appendix A).

## 3. Discussion

This study showed that *NUCB2* expression in the placenta might regulate trophoblast differentiation and fusion as pregnancy progresses. *NUCB2* mRNA and protein expression progressively increases with temporal progression of spontaneous primary trophoblast syncytization and forskolin-induced BeWo cell fusion. High concentrations of Nesfatin-1 inhibited the differentiation and fusion of primary trophoblast and Forskolin-induced BeWo cells. The differentiation and fusion of BeWo cells were also inhibited after *NUCB2* knockdown. The RNA-seq results indicated that *NUCB2* might be involved in the differentiation and fusion process through the *EGFR-PLCG1-CAMK4* signaling pathway. The present study provided new evidence that *NUCB2*/Nesfatin-1 was involved in the differentiation and fusion process of trophoblast cells to some extent.

E-cadherin is a key adhesion molecule associated with syncytialization. It has been reported that intercellular adhesion was increased during the syncytialization of trophoblasts, whereas the expression of E-cadherin was significantly reduced in trophoblasts from fused cells [25,26]. Previous studies have found that the rat placental labyrinth differentiates into a first layer adjacent to the circulating trophoblasts in the maternal bloodstream, an intermediate second layer, and a third layer in contact with the endothelial basement membrane and mesenchyme of the fetal vasculature. From day 14 to day 18 of gestation, trophoblastic cells undergo extensive cell fusion, especially from day 14 to day 16, when the structure of the trabeculae is altered and there is a decrease in the number of undifferentiated cells, the platelet arrangement is more pronounced, and the trophoblastic layer becomes contiguous [27]. A significant decrease in E-cadherin mRNA levels from day 14.5 to day 17.5 of gestation was found in this study, which implies that the level of placental syncytialization was increased from day 14.5 to day 17.5 of gestation. This decrease in E-cadherin mRNA levels was followed by a significant increase in the mRNA levels of E-cadherin on day 19.5 of gestation. This result suggests a trend of increasing and then decreasing placental syncytialization levels in rats from day 14.5 to day 19.5 of gestation. Previous studies found that placental *NUCB2* mRNA levels decreased significantly as gestation progressed, with a decrease in *NUCB2* expression observed from day 16 of gestation onwards, reaching a minimum on day 21. At the same time, the Western blot analysis showed a significant decrease in *NUCB2* protein on day 21 of gestation [28,29]. It was also found that placental cells were in a state of extensive differentiation and fusion from days 14 to 18 of gestation, whereas the results of Western blot in these studies only showed a significant decrease in *NUCB2* protein on day 21 of gestation, and the trends at other timepoints were not described in detail [28]. New evidence reveals that *NUCB2* expression in the placenta during pregnancy may be a dynamic process, with *NUCB2* mRNA levels increasing and then decreasing from day 14.5 to day 19.5 of gestation. This is consistent with the trend of syncytialization levels, suggesting a link between *NUCB2* levels and trophoblast syncytialization.

*NUCB2* and HCG-β expression in mRNA and protein expression levels were measured in human primary trophoblast cells isolated from healthy term placentas and in a Forskolin-induced BeWo cell fusion model at various time points to investigate the connection between *NUCB2* and trophoblast syncytialization. HCG-β is an important pregnancy hormone, as it is synthesized by STBs and is considered a biochemical marker of syncytialization. The expression of HCG-β was measured to demonstrate the success of the spontaneous trophoblast syncytialization model and to establish a model of Forskolin-induced differentiation [14]. The results of the present study suggest that the expression of *NUCB2* gradually increased with the spontaneous syncytialization process, which was consistent with the results of previous studies [28]. Meanwhile, the *NUCB2* mRNA and protein levels in primary trophoblast cells considerably increased after 24 h of spontaneous syncytialization culture. Also, previous studies found that *NUCB2*/Nesfatin-1 was highly expressed in the STBs of human floating villi throughout gestation, and the spontaneous syncytialization model mimics this process of CTBs differentiating into STBs [14]. It was demonstrated that BeWo cells could fuse into multinucleated cells and form syncytia upon stimulation with Forskolin [30,31]. This could be observed via immunostaining of E-cadherin [25]. A previous study [19] examined genes and isozymes involved in Forskolin-induced BeWo cell fusion using RNA sequencing (RNA-Seq). *NUCB2* was significantly upregulated at 48 h after Forskolin induction, which was consistent with our findings, although we found no significant changes in *NUCB2* levels at 24 h after Forskolin induction, neither at the mRNA level nor the protein level, in BeWo cells [19]. The immunofluorescence data presented in this study also supported the finding that *NUCB2* was mostly expressed in cells in which STB fusion had taken place. In summary, in primary trophoblast cells and in BeWo cells, *NUCB2* expression increased with an increase in the degree of trophoblast differentiation and fusion.

Exogenous Nesfatin-1, which serves as the active center of *NUCB2* [32], was applied at various concentrations to primary trophoblast cells and BeWo cells to further investigate the impact of *NUCB2*/Nesfatin-1 on trophoblast differentiation and fusion. The results indicated that high concentrations of Nesfatin-1 might inhibit trophoblast differentiation and fusion, as indicated by the fusion rate and levels of syncytial markers. Enhanced trophoblast differentiation, hCG hormone expression, and placental alkaline phosphatase (PLAP) expression were all associated with syncytialization [21,33]. Previous studies have demonstrated that the endogenous retroviral genes ERVW-1 and ERVFRD-1 play important roles in promoting trophoblast fusion [34,35]. ERVW-1 [36] and sERVFRD-1 [37] have been found to be present on the surface of trophoblast-derived extracellular vesicles with immunosuppressive activity. Extracellular vesicles (EVs) are small membrane-encapsulated particles released by different types of cells, such as monocytes, endothelial cells, platelets, tumor cells, and chorionic STBs [38], which are shed into the chorionic interstitium and enter the maternal circulation via the continuous turnover of the STB layers during the progression of pregnancy [39]. Glial cell deletion-1 (GCM1), a transcription factor, is highly expressed in human syncytiotrophoblasts and has been suggested to be a major regulator of syncytiotrophoblast formation [40]. High concentrations of Nesfatin-1 might inhibit trophoblast differentiation and fusion, as evidenced by a decrease in the fusion rate, a reduction in PLAP activity, and a decrease in several differentiation and fusion indicators (ERVW-1, ERVFRD, GCM1, and HCG-β). Nesfatin-1 levels in cord blood were shown to be considerably lower in pregnant women with fetal weights that were large for the gestational age (LGA) compared to those appropriate for the gestational age (AGA) in a previous investigation [41]. In a different study, it was discovered that patients with intrauterine growth retardation (IUGR) had significantly higher levels of maternal serum Nesfatin-1 compared to patients without IUGR. Elevated maternal Nesfatin-1 levels were negatively correlated with fetal weights, indicating that high maternal Nesfatin-1 levels inhibited intrauterine fetal growth. In another study, plasma Nesfatin-1 levels were found to be higher in children with SGA than in those with AGA. Cellular trophoblasts from IUGR placentas were found to have significantly lower cell fusion indices and nuclei per syncytiotrophoblast in vitro than those from placentas of normal pregnancies, while both human ERVW-1 and ERVFRD were found to exhibit downregulation in the IUGR group [42]. Another study found that IUGR trophoblast cells exhibited increased differentiation, such as increased syncytiotrophoblastization and increased β-hCG production, compared to trophoblast cells from uncomplicated pregnancies [43]. These studies suggested that Nesfatin-1 might affect the fusion process of trophoblastic differentiation and, subsequently, fetal growth and development. 

The *NUCB2* gene was knocked out in the BeWo cell line to further investigate the role of the *NUCB2* gene in trophoblast differentiation and fusion. The BeWo cell line was chosen over primary trophoblast cells primarily because the latter could not proliferate in a cell culture and a stable knockout cell line could not be created using them. The *NUCB2* knockdown group displayed varying degrees of PLAP activity reduction compared to the control, transfected BeWo cells, and wild-type BeWo cells treated with the same concentration of Forskolin. Markers of syncytialization like HCG-β, ERVW-1, and ERVFRD were also downregulated, and a decline in the fusion index was confirmed morphologically. These results suggest that *NUCB2* is important in the process of trophoblast differentiation and fusion and that high doses of Nesfatin-1 inhibit this process, while the knockdown of *NUCB2* similarly inhibits trophoblast differentiation and fusion. In this study and many others, *NUCB2* is highly expressed in syncytiotrophoblast cells, suggesting that Nesfatin-1 plays a role in maintaining the balance between syncytiotrophoblasts and cytotrophoblasts. To explore the mechanisms by which the *NUCB2* gene regulates trophoblast differentiation and fusion, RNA-seq analysis was conducted using lentiviral control and lentiviral knockout groups. The findings were in line with previous investigations, and the KEGG results showed that *NUCB2* knockdown decreased the expression of genes related to the calcium signaling pathway, such as genes involved in the EGFR-PLCG1-CAMK4 pathway. Previous investigations found that during the naturally occurring formation of syncytiotrophoblast cells in isolated human expired placentas, hCG secretion and Ca^2+^ uptake by trophoblast cells increased gradually with the number of days in culture [44]. EGFR is highly abundant in the placenta and plays an important role in key cellular processes such as cell proliferation, differentiation, invasion, and fusion [45]. A previous study found that in utero exposure to bisphenol S (BPS) interfered with epidermal growth factor (EGF)-dependent activation of EGFR, thereby reducing the fusion of human placental cells [44]. Evidence suggests that EGF-induced phosphorylation and activation of PLCG1 is required for EGF receptor-mediated cell motility and that PLCG1 is closely associated with trophoblast cell proliferation and motility [46]. Knockdown of CAMK4 was found to inhibit the proliferation and migratory ability of HTR-8/SVneo cells [47,48]. CAMK4 is involved in a variety of biological processes such as trophoblast cell differentiation, invasion, and embryonic development, which suggests that CAMK4 has an effect on trophoblast cell differentiation to some extent. It has been hypothesized that calcium signaling channels (e.g., epidermal growth factor receptor-PLCG1-CAMK4) might be involved in trophoblast cell differentiation and fusion. The mRNA and protein levels of the epidermal growth factor receptor (EGFR)-PLCG1-CAMK4 gene in the lentiviral control and lentiviral knockout groups were further examined, and it was found that the mRNA level of the epidermal growth factor receptor (EGFR)-PLCG1-CAMK4 gene in the *NUCB2* knockout group was significantly reduced compared to that in the lentiviral control group. The total protein levels of epidermal growth factor receptor (EGFR) and PLCG1 were significantly reduced, and the total protein level and phosphorylated protein level of CAMK4 also exhibited a tendency of being reduced.

## 4. Materials and Methods

### 4.1. Animal Models

Sprague Dawley rats (purchased from Beijing Viton Lever Laboratory Animal Technology Co., Ltd., Beijing, China) were housed under standard environmental conditions (12 h dark–light cycle) with appropriate temperature and humidity. These rats had free access to food and water. Virgin female rats were mated with male rats in a 2:1 co-cage, and the vaginal plug was observed and designated as E0.5. All experiments were approved by the Animal Ethics Committee of the Capital Medical University under protocol number AEEI-2017-101.

### 4.2. Isolation of Primary Human Trophoblast Cells

This study was approved by the Ethics Committee of the Capital Medical University (2018SY04). Placentas from healthy women aged 20–35 who delivered at full term were collected from the Capital Medical University-affiliated Fuxing Hospital. According to a previously published study, primary human trophoblast (PHT) cells were isolated from human-term placentas for subsequent studies [49,50]. The placentas were manually cut into pieces and digested three times in DMEM (HyClone, Logan, UT, USA) with 0.125% trypsin (Gibco, 25200-072, New York, NY, USA) and 0.02% DNase I (Sigma Aldrich, St. Louis, MO, USA, DN25-100 mg). The obtained supernatant was centrifuged at 2400 rpm for 10 min to obtain a precipitate containing the cell pellet, which was then separated via density gradient centrifugation using Percoll (Cytiva, 17089101, Washington, DC, USA). PHT cells were purified in a 30–50% density gradient of Percoll. The obtained PHT cells were cultured in DMEM containing 10% fetal bovine serum (FBS; Excell Bio, FSP500, Shanghai, China), 100 units/mL penicillin, and 100 units/mL streptomycin in 6-well plates (2 × 10^6^/plate), and incubated in a 5% CO_2_ air incubator at 37 °C. CK7 immunofluorescence was used to identify the extracted cells as trophoblast cells, which showed that approximately 91% of the extracted cells had a positive signal for CK7 (Appendix A). PHT cells were cultured for up to 24 h with human recombinant Nesfatin-1 (4, 20, 100, and 500 ng/mL) (Novus, Cat. No.: CR25, Lone Tree, CO, USA) in a serum-free medium for 4 h. PHT cells were then cultured in a medium containing 10% fetal bovine serum for up to 72 h and collected for subsequent experiments.

### 4.3. BeWo Cell Culture and Fusion Assay

The human choriocarcinoma cell line BeWo was purchased from the National Infrastructure of Cell Line Resource (Beijing, China) and cultured in F-12 (Gibco, C11765500BT) containing 10% fetal bovine serum, 100 units/mL penicillin, and 100 units/mL streptomycin in a 5% CO_2_ incubator at 37 °C. After being inoculated in 6-well plates (5 × 10^5^ cells/mL), BeWo cells were treated with human-derived recombinant Nesfatin-1 (4, 20, 100, and 500 ng/mL) (Novus, Cat. No.: CR25, CO, USA) in a serum-free medium at 37 °C for 4 h to determine the effect of Nesfatin-1, followed by treatment with 50 μM Forskolin (FSK. Targetmol, T2939, Shanghai, China) for 48 h to induce cell fusion. Afterwards, these cells were collected for subsequent experiments.

### 4.4. Lentiviral Vector Infection for NUCB2 Knockdown

The human *NUCB2* siRNA lentiviral vector construction and packaging services were provided by Hanbio Co., Ltd. (Shanghai, China). The interference target design and primers were as follows: siRNA1: CCACAGATTTAGATATGCTAA; siRNA2: CCAGGAAGCAAAGATCAACTA; and siRNA3: GCTGGAATATCATCAGGTCAT. Double-stranded oligonucleotides were inserted between the pHBLV-U6-MCS-CMV-ZsGreen-PGK-PURO plasmid BamHI/EcoRI restriction sites, and the ligated plasmids were transformed into E. coli DH-5α competent cells for plasmid amplification. Plasmids from the positive colonies were verified via RT-PCR and DNA sequencing. BeWo cells were prepared in suspension, inoculated into 6-well plates with 1 × 10^5^ cells per well, and incubated for 24 h. When the cell density reached about 50%, a suitable volume of lentiviral infection reagent was added to the medium with the multiplicity of infection (MOI) of 20. After 24 h, the virus-containing medium was replaced with a fresh complete medium. The effectiveness of transfection was determined based on green fluorescent protein (GFP) expression at 48 h post-transfection. After 72 h of transfection, BeWo cells were cultured in a fresh complete medium containing 5 μg/mL puromycin to establish a stable transfected cell line for all subsequent lentiviral cell experiments. The established stable virus-transfected BeWo cell line was cultured in a fresh complete medium containing 50 μM Forskolin for 48 h, and RNA samples and protein samples were collected for subsequent experiments.

### 4.5. Placental Alkaline Phosphatase (PLAP) Activity Assay

Protein was extracted from PHT and BeWo cells using ice-cold RIPA lysate (Biyuntian Institute of Biotechnology, Shanghai, China). PLAP activity in the supernatant of cell lysates (PHT cells: 30 μg protein/well and BeWo cells: 40 μg protein/well) was detected using a Placental Alkaline Phosphatase Assay Kit (Biyuntian Institute of Biotechnology, Shanghai, China), and the procedure was performed according to the kit’s instructions.

### 4.6. Total RNA Extraction and Real-Time PCR

Total RNA from cells and placental tissue was extracted using TRIzol reagent according to the manufacturer’s instructions (Invitrogen, Carlsbad, CA, USA). The concentration and purity of total RNA were measured using a spectrophotometer (Biotek Epoch, Winooski, VT, USA). cDNA synthesis was performed using the RevertAid First Strand cDNA Synthesis Kit (Thermo Scientific, Waltham, MA, USA) according to the manufacturer’s protocol. Real-time PCR was then performed using the KAPA SYBR FAST qPCR Kit (Kapa Biosystems, Boston, MA, USA). A 20 μL reaction system consisting of 10 μL of master mix (2×), 0.4 μL of each primer (10 μM), 1 μL of cDNA, and 8.2 μL of PCR grade water was used. The primers for *NUCB2*, ERVW-1, ERVFRD, HCG-β, GCM1, EGFR, PLCG1, PLCG2, KRAS, CAMK4, GAPDH, and β-actin are detailed in Appendix A. Relative mRNA expression was calculated using the 2^−ΔΔCt^ method.

### 4.7. Western Blotting Analysis

Proteins from cells or tissues were extracted using RIPA lysate containing phosphatase inhibitors and PMSF denatured at 100 °C for 5 min, and 30 μg of denatured proteins per well was subjected to SDS-PAGE. Afterwards, membranes were closed with 5% skim milk for 2 h at room temperature and incubated with the indicated antibodies at 4 °C (Appendix A). Subsequently, membranes were incubated with the corresponding horseradish peroxidase (HRP)-conjugated secondary antibodies (Appendix A). Signals were visualized using an electrochemiluminescence (ECL) kit (Merck Millipore, Billerica, MA, USA, WBKLS0100), and then exposed using an image system Fusion FX (Vilber Lourmat Co., Ltd., Paris, France) and analyzed with image analysis software (VILBER LOURMAT Fusion FX7, Paris, France). The experiment was repeated independently three times.

### 4.8. Immunofluorescence Staining and Confocal Microscopy

The paraffin sections of the rat placentas were deparaffinized with xylene three times for 15 min each time, and then dehydrated with 100% alcohol, 95% alcohol, 85% alcohol, and 70% alcohol for 5 min, followed by antigen repair and removal of catalase with 3% hydrogen peroxide. For immunofluorescence staining of PHT cells and BeWo cells, cells were inoculated onto confocal disks, fixed with 4% paraformaldehyde for 15 min, and then permeabilized with 0.25% Triton X-100 (Sigma-Aldrich) PBS for 10 min. After the paraffin sections and cells were blocked with 10% goat serum for 1 h, these sections and cells were incubated overnight at 4 °C with E-cadherin mouse monoclonal antibody (1:50 dilution) and *NUCB2* rabbit polyclonal antibody (1:500 dilution). Afterwards, all operations were performed at room temperature and protected from light. The paraffin sections and cells were rinsed 3 times with PBS, and then goat anti-mouse Alexa Flour 488 antibody (Abcam, 1:500 dilution, Cambridge, UK) and goat anti-rabbit Alexa Flour 562 antibody (Abcam, 1:500 dilution, Cambridge, UK) were added and incubated for 1.5 h. Staining was performed with 300 μL of DAPI (DAPI; solaba) for 10 min at 4°C, rinsed three times with PBS, and then photographed using a confocal microscope.

### 4.9. RNA Sequencing

Total RNA was extracted from 2 × 10^6^ BeWo cells after 48 h of Forskolin or DMSO treatment using TRIzol reagent, with three replicate wells per sample, and dissolved in 40 μL of DEPC-H2O. RNA was assayed for purity and concentration using NanoDrop 2000 (illumina, San Diego, CA, USA), and then Agilent 2100/4200 (illumina, San Diego, CA, USA) was used to assess the RNA samples for integrity and quantity. The first-strand cDNA was reverse-transcribed with fragmented RNA and dNTP (dATP, dTTP, dCTP, and dGTP), followed by second-strand cDNA synthesis. The remaining double-stranded cDNA was converted to blunt ends via exonuclease/polymerase activity. After adenylation of the 3′ end of the DNA fragment, sequencing connectors were ligated to the cDNA and the library fragment was purified. The template was enriched with PCR, the PCR products were filtered to obtain the final library, and then the samples were sequenced by Illumina. Differential expression was analyzed using EdgeR. The resulting *p*-values were adjusted using the method of Benjamini and Hochberg to control the false discovery rate. Genes with |log2 (fold change)| > 1 and q-values < for 0.05 were classified as differentially expressed genes. GO and KEGG enrichment analyses of differentially expressed gene sets were performed using the topG method.

### 4.10. Statistical Analysis

All statistical analyses were performed using the SPSS 23.0 package (IBM). All data were based on at least three independent experiments, and the results are reported as mean ± SD. Differences between groups were analyzed using one-way ANOVA for group differences (normally distributed data), and we also used the post hoc Dunnett *t*-test to ascertain the differences between groups when the ANOVA results showed *p* < 0.05. *p* < 0.05 was considered statistically different between the two groups.

## 5. Conclusions

In summary, *NUCB2*/Nesfatin-1 is important in trophoblast cell syncytialization. The level of trophoblast syncytialization was found to decrease after the knockdown of *NUCB2*. *NUCB2* may regulate trophoblast differentiation and fusion through the EGFR-PLCG1-CAMK4 signaling pathway. We also discovered an interesting phenomenon that Nesfatin-1 produces an inhibitory effect when cells are given a specific concentration of exogenous Nesfatin-1, but the exact mechanism underlying this phenomenon remains to be further explored.

## Figures and Tables

**Figure 1 ijms-25-01925-f001:**
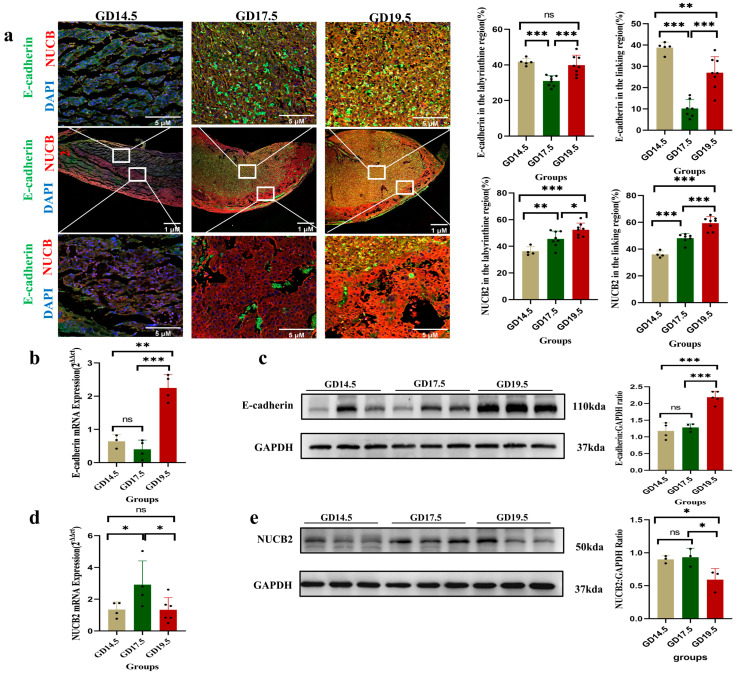
Expression and localization of Nuclear Binding Protein 2 (*NUCB2*) and E-cadherin in *rat* placentas at different gestation stages. (**a**) Immunofluorescence of *rat* placentas is shown on GD 14.5, GD 17.5, and GD 19.5 days. Representative images of the placenta are shown (*n* = 3). Immunofluorescence staining was performed with antibodies against *NUCB2* (red) and E-cadherin (green), and DAPI (blue) was used to stain cell nuclei. Changes in E-cadherin and *NUCB2* fluorescence areas in the labyrinthine and connective regions between 14.5 and 19.5 days of gestation were statistically analyzed using the ImageJ software V 1.8.0. The scale bar represents 1 μM and 5 μM. (**b**) The expression of E-cadherin mRNA in *rat* placentas at different gestation stages was studied using RT-PCR. (**c**) Western blotting was used to study the expression of E-cadherin protein in the placenta of *rat*s at different gestational stages. (**d**) The expression of *NUCB2* gene in *rat* placentas at different gestation stages was studied using RT-PCR. (**e**) Western blotting was used to study the expression of *NUCB2* protein in the placenta of *rat*s at different gestational stages. One-way ANOVA followed by Tukey’s test was performed (ns stands for not significant, * *p* < 0.05, ** *p* < 0.01, and *** *p* < 0.001).

**Figure 2 ijms-25-01925-f002:**
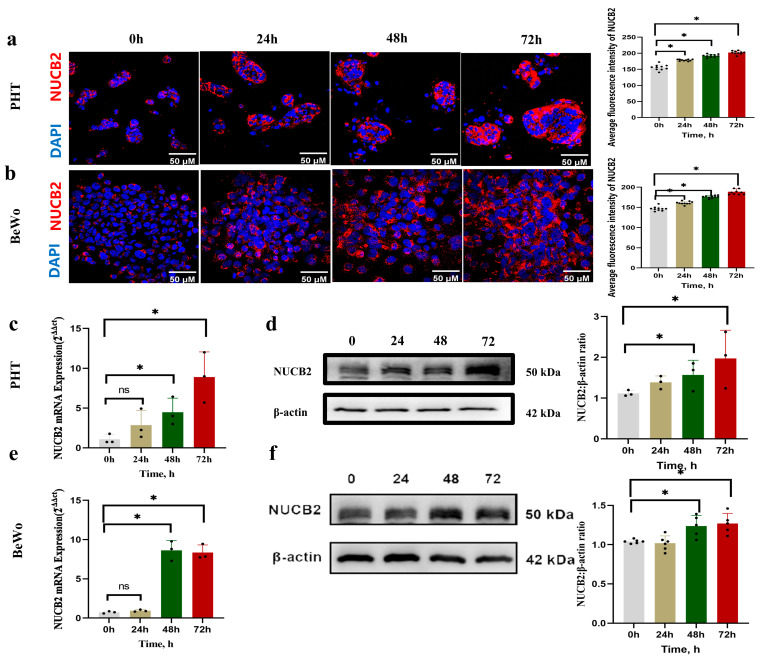
*NUCB2* increases during trophoblast syncytialization. (**a**) Representative immunofluorescence images depicting changes in *NUCB2* at different time points of primary trophoblast syncytialization. The intensity of *NUCB2* at different time points of primary trophoblast syncytialization was analyzed using the ImageJ software, and the scale bar represents 50 μM. Cells were stained with anti-*NUCB2* (red) or DAPI (blue). (**b**) Representative immunofluorescence images depicting changes in *NUCB2* at different time points of Forskolin-induced differentiation and fusion of BeWo cells. The intensity of *NUCB2* at different time points of Forskolin-induced differentiation and fusion of BeWo cells was analyzed using the ImageJ software, and the scale bar represents 50 μM. Cells were stained with anti-*NUCB2* (red) or DAPI (blue). (**c**) The expression of *NUCB2* mRNA at different time points during primary trophoblast syncytialization. (**d**) Western blot detection of protein expression levels of *NUCB2* at different time points during trophoblast syncytialization. (**e**) The expression of *NUCB2* mRNA at different time points of Forskolin-induced differentiation and fusion of BeWo cells. (**f**) Western blot detection of protein expression levels of *NUCB2* at different time points of Forskolin-induced differentiation and fusion of BeWo cells. Data are presented as the mean ± SD of three independent experiments. One-way ANOVA followed by Tukey’s test was performed (ns stands for not significant, * *p* < 0.05,).

**Figure 3 ijms-25-01925-f003:**
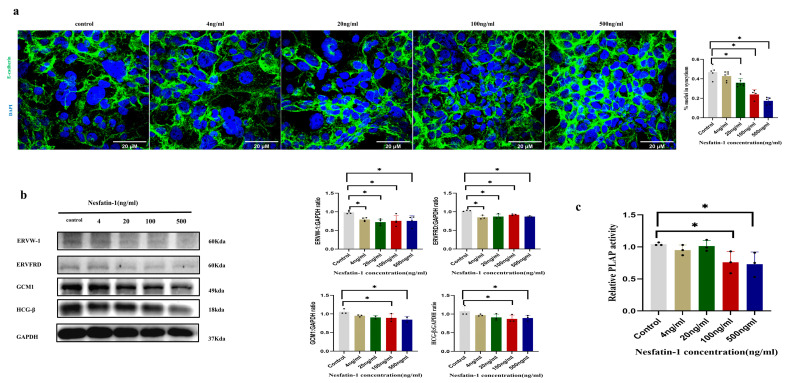
Exogenous Nesfatin-1 inhibits Forskolin-induced differentiation and fusion of BeWo cells. (**a**) Representative immunofluorescence images depicting the effect of exogenous Nesfatin-1 on differentiation and fusion in Forskolin-induced BeWo cells; cells were stained with anti-E-cadherin (green) or DAPI (blue). The scale bar represents 20 μM. The degree of cell fusion was quantified based on the fusion index. The fusion index was calculated by dividing the total number of fused cells (cells with at least 2 nuclei) in each field of view by the total number of cells in the field of view. (**b**) The effect of exogenous Nesfatin-1 on protein expression levels of genes (including ERVW-1, ERVFRD, GCM1, and HCG-β) in Forskolin-induced BeWo cells. (**c**) The effect of exogenous Nesfatin-1 on PLAP activity in Forskolin-induced BeWo cells. Data are presented as the mean ± SD of three independent experiments. One-way ANOVA followed by Tukey’s test was performed (* *p* < 0.05).

**Figure 4 ijms-25-01925-f004:**
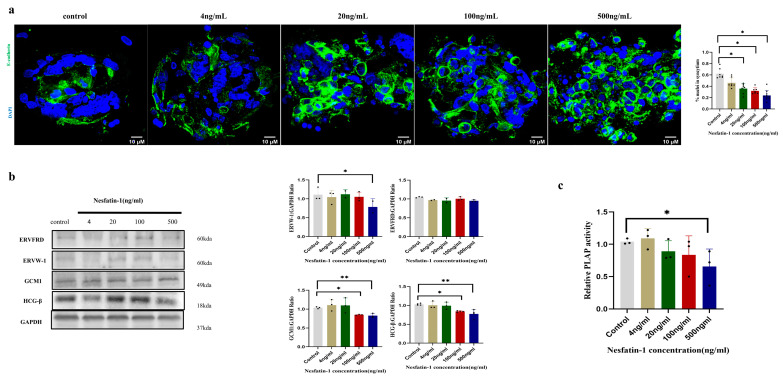
Exogenous Nesfatin-1 inhibits the differentiated fusion of primary trophoblast cells. (**a**) Representative immunofluorescence images depicting the effect of exogenous Nesfatin-1 on primary trophoblast cell syncytialization; cells were stained with anti-E-cadherin (green) or DAPI (blue). The scale bar represents 10 μM. The degree of cell fusion was quantified based on the fusion index. The fusion index was calculated by dividing the total number of fused cells (cells with at least 2 nuclei) in each field of view by the total number of cells in the field of view. (**b**) The effect of exogenous Nesfatin-1 on protein expression levels of genes (including ERVW-1, ERVFRD, GCM1, and HCG-β) during primary trophoblast syncytialization. (**c**) The effect of exogenous Nesfatin-1 on PLAP activity during primary trophoblast cell syncytialization. Data are presented as the mean ± SD of three independent experiments. One-way ANOVA followed by Tukey’s test was performed (* *p* < 0.05, ** *p* < 0.01).

**Figure 5 ijms-25-01925-f005:**
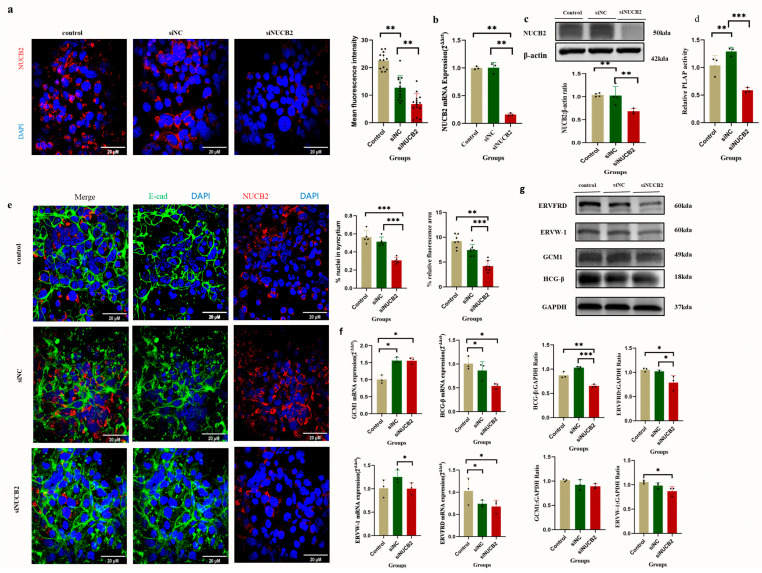
Effect of *NUCB2* knockdown on Forskolin-induced differentiation and fusion of BeWo cells. (**a**) Representative immunofluorescence images depicting the knockdown of endogenous *NUCB2* in BeWo cells. The intensity of *NUCB2* in *NUCB2*-knockdown BeWo cells was analyzed using ImageJ software and the scale represents 20 μM. Cells were stained with anti-*NUCB2* (red) or DAPI (blue). (**b**) Endogenous *NUCB2* mRNA levels in *NUCB2*-knockdown BeWo cells; (**c**) indicates endogenous *NUCB2* protein levels in *NUCB2*-knockdown BeWo cells. (**d**) Indicates the effect of *NUCB2* knockdown on Forskolin-induced differentiation and fusion of BeWo trophoblast cells as measured by PLAP activity. (**e**) Representative immunofluorescence images depicting the effect of the knockdown of endogenous *NUCB2* in BeWo cells on the Forskolin-induced fusion rate of BeWo trophoblast cells. The intensity of *NUCB2* in *NUCB2*- knockdown BeWo cells was analyzed using the ImageJ software. The degree of cell fusion was quantified based on the fusion index. The fusion index was calculated by dividing the total number of fused cells (cells with at least 2 nuclei) in each field of view by the total number of cells in the field of view. Cells were stained with anti-*NUCB2* (red) or DAPI (blue). (**f**) The effect of knockdown of endogenous *NUCB2* in BeWo cells on the mRNA levels of genes (including *ERVW-1, ERVFRD, GCM1*, and HCG-β) in Forskolin-induced BeWo trophoblast cells. (**g**) The effect of knockdown of endogenous *NUCB2* on the Forskolin-induced protein expression levels of trophoblast differentiation and fusion genes (including *ERVW-1, ERVFRD, GCM1*, and HCG-β) in BeWo trophoblast cells. Data are presented as the mean ± SD of three independent experiments. One-way ANOVA followed by Tukey’s test was performed (* *p* < 0.05, ** *p* < 0.01, and *** *p* < 0.001).

**Figure 6 ijms-25-01925-f006:**
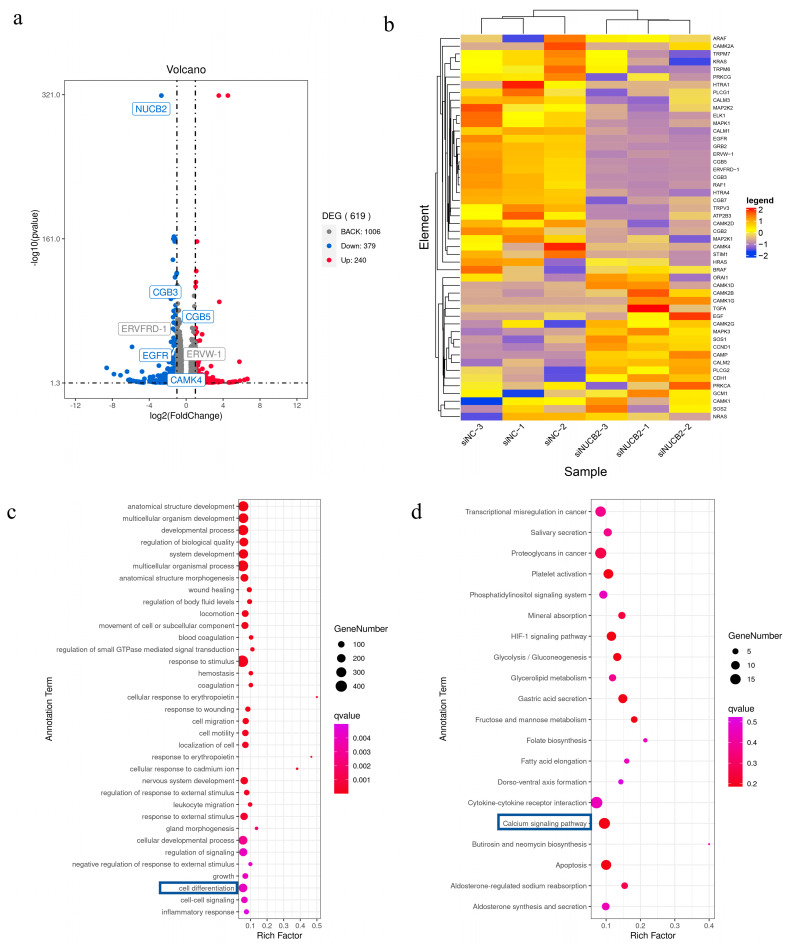
*NUCB2* plays a role in influencing trophoblast differentiation and fusion through the calcium signaling pathway. (**a**) Volcano plot showing differentially expressed genes in BeWo cells in the lentiviral control and *NUCB2* knockdown groups in Forskolin-induced BeWo cells. (**b**) Heatmap showing differentially expressed genes in BeWo cells in the lentiviral control and *NUCB2* knockdown low groups after being treated with Forskolin for 48 h in BeWo cells. (**c**) Down-regulated functional enrichment in lentiviral control and *NUCB2* knockdown low expression groups. The blue box represents the “cell differentiation” function, emphasizing the concern. (**d**) Down-regulated KEGG signaling pathway enrichment pathway in lentiviral control and *NUCB2* knockdown low-expression groups. The blue box represents the “calcium signaling pathway” function, emphasizing the concern.

**Figure 7 ijms-25-01925-f007:**
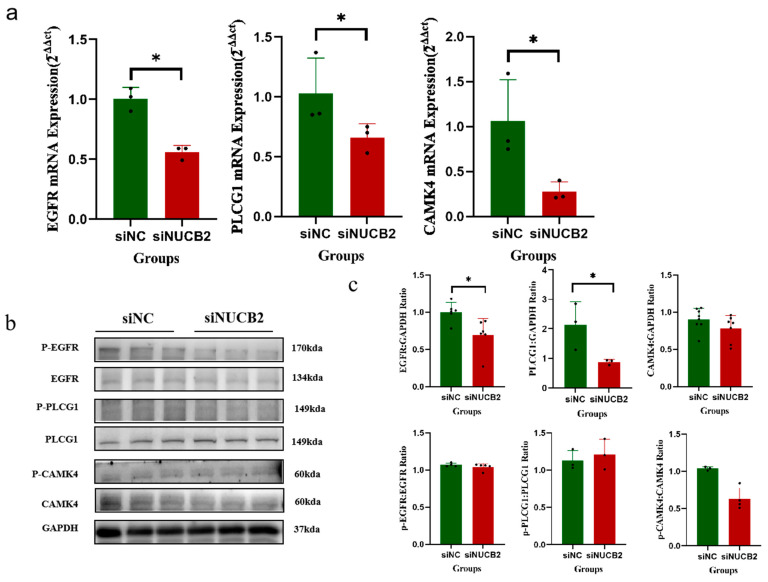
Changes in factors associated with the epidermal growth factor receptor (EGFR)-phospholipase C gamma 1 (PLCG1)-calmodulin-dependent protein kinase IV (CAMK4) signaling pathway after *NUCB2* knockdown. (**a**) The effect of endogenous BeWo-cell *NUCB2* knockdown in BeWo cells on EGFR, PLCG1, and CAMK4 mRNA levels during Forskolin-induced BeWo trophoblast differentiation and fusion. (**b**,**c**) The effect of endogenous BeWo-cell *NUCB2* knockdown in BeWo cells on the levels of EGFR, PLCG1, and CAMK4 proteins during Forskolin-induced BeWo trophoblast differentiation and fusion. Data are presented as the mean ± SD of three independent experiments. One-way ANOVA followed by Tukey’s test was performed (* *p* < 0.05).

## Data Availability

All data generated or analyzed during this study are included in this published article and its Appendix A.

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
