# Peer review of "Nuclear Binding Protein 2/Nesfatin-1 Affects Trophoblast Cell Fusion during Placental Development via the EGFR-PLCG1-CAMK4 Pathway"

_ijms, 2024, doi:10.3390/ijms25031925_

Round 1

Reviewer 1 Report

Comments and Suggestions for Authors

I reviewed the research article titled “NUCB2/Nesfatin-1 Contributes to Trophoblast Cell Fusion in Placenta Development via the EGFR-PLCG1-CAMK4 Pathway”. Although the study doesn’t bring any excitement to the readership, it might contribute to the literature. NUCB2/Nesfatin-1 mainly involves energy homeostasis and the placenta is an actively developing tissue during pregnancy, requiring high energy demand. The obvious fact is that there should be some dependency.

The study includes several experiments to demonstrate that NUCB2/Nesfatin-1 involves in syncytialization of trophoblasts and acts through the EGFR-PLCG1-CAMK4 Pathway. There are a considerable number of language flaws and science/technical flaws throughout the manuscript. Clarity and readability are lacking in some places. Some of them are as follows.

1) I would avoid declarative sentences in titles and subtitles since declarative sentences or phrases in scientific article titles tend to overemphasize a conclusion.

2) The abstract requires language correction to some extent. Avoid using too many first-person nouns. Since NUCB2 is the precursor of Nesfatin-1, using only NUCB2 instead of NUCB2/Nesfatin-1 would be better when the gene expression is considered.

3) The introduction section seems alright. But there is nothing to introduce the EGFR-PLCG1-CAMK4 Pathway, connecting to NUCB2 or trophoblast syncytialization.

4) In materials and methods, the clarity is lacking in lines 408-409, 414-419, 436-438, 446-447 (write from what tissue/cells), 452-453 (mention the target genes), 467-469 (write paraffin section of what), 479-480 (anti-attenuation blocking?) and so on. Authors are advised to reread to enhance the entire method section.

5) In the results section, the clarity and readability need improvements in lines 84-87. In Figure 1a, it looks like all rows are merged. Why is the middle row labeled as DAPI (counterstain)? Immunohistochemistry picture can be improved in quality. Labeling needs to be corrected. How is the fluorescence intensity quantified? Figure 2a looks like merged images and the labeling needs to be corrected accordingly. Also, it has two same bar graphs. How is it quantified? Are Figure 3a images under same magnification? Are bar lines shown? In lines 153-155, how long are they treated with Nesfatin-1? Do you determine the percentage of fusion rate by E-cadherin? Please address the similar concern for Figure 4a. There is no description for section 2.6 (lines 235-236).

6) In the discussion section, in line 264, what is the maternal blood trophoblast? In lines 287-289, the clarity is lacking. For lines 356-359, there is nothing in the results section, connecting the EGFR-PLCG1-CAMK4 pathway and related genes. Overall, the discussion section requires some improvements.            

Comments on the Quality of English Language

Language improvement is required to some extent to enhance the clarity and readability of the manuscript. 

Author Response

Reviewers 1

I reviewed the research article titled “NUCB2/Nesfatin-1 Contributes to Trophoblast Cell Fusion in Placenta Development via the EGFR-PLCG1-CAMK4 Pathway”. Although the study doesn’t bring any excitement to the readership, it might contribute to the literature. NUCB2/Nesfatin-1 mainly involves energy homeostasis and the placenta is an actively developing tissue during pregnancy, requiring high energy demand. The obvious fact is that there should be some dependency.

The study includes several experiments to demonstrate that NUCB2/Nesfatin-1 involves in syncytialization of trophoblasts and acts through the EGFR-PLCG1-CAMK4 Pathway. There are a considerable number of language flaws and science/technical flaws throughout the manuscript. Clarity and readability are lacking in some places. Some of them are as follows.

  • I would avoid declarative sentences in titles and subtitles since declarative sentences or phrases in scientific article titles tend to overemphasize a conclusion.

Response: Many thanks for your careful check. We have revised the title and subtitle of the article, as described in the revised manuscript.

  • The abstract requires language correction to some extent. Avoid using too many first-person nouns. Since NUCB2 is the precursor of Nesfatin-1, using only NUCB2 instead of NUCB2/Nesfatin-1 would be better when the gene expression is considered.

Response: Many thanks for this comment. We have modified the use of the first person in the abstract and modified NUCB2/Nesfatin-1 to NUCB2, as described in the revised manuscript.

  • The introduction section seems alright. But there is nothing to introduce the EGFR-PLCG1-CAMK4 Pathway, connecting to NUCB2 or trophoblast syncytialization.

Response: Many thanks for this comment. We have added the EGFR-PLCG1-CAMK4 pathway with NUCB2 or trophoblast syncytization to the Introduction, which can be found on lines 67-74 of the revised manuscript.

  • In materials and methods, the clarity is lacking in lines 408-409, 414-419, 436-438, 446-447 (write from what tissue/cells), 452-453 (mention the target genes), 467-469 (write paraffin section of what), 479-480 (anti-attenuation blocking?) and so on. Authors are advised to reread to enhance the entire method section.

Response: Many thanks for this comment. We have revised the issues in the Materials and Methods section accordingly, and the changes can be found in the revised manuscript.

  • In the results section, the clarity and readability need improvements in lines 84-87. In Figure 1a, it looks like all rows are merged. Why is the middle row labeled as DAPI (counterstain)? Immunohistochemistry picture can be improved in quality. Labeling needs to be corrected. How is the fluorescence intensity quantified? Figure 2a looks like merged images and the labeling needs to be corrected accordingly. Also, it has two same bar graphs. How is it quantified? Are Figure 3a images under same magnification? Are bar lines shown? In lines 153-155, how long are they treated with Nesfatin-1? Do you determine the percentage of fusion rate by E-cadherin? Please address the similar concern for Figure 4a. There is no description for section 2.6 (lines 235-236).

Response: Many thanks for your careful check. We have rewritten lines 84-87 of the manuscript to read as follows: Using double immunofluorescence labeling, we observed consistent changes in the fluorescence area of E-cadherin and NUCB2 in the labyrinthine and connective areas from gestational days 14.5 to 19.5. There was a significant decrease on gestational day 17.5 compared to that on gestational day 14.5 (P<0.001), and a significant increase on gestational day 19.5 compared to that on gestational day 17.5 (P<0.001). Each image in Figure 1a is trichrome stained for E-cadherin, NUCB2, and DAPI, where the first and third rows are enlarged areas of the placental junction and labyrinthine regions, respectively, which we have re-marked. In addition, we quantified the fluorescence area of each region using ImageJ software. We changed the labeling of Fig. 2a and changed the incorrect bars. The brightness of NUCB2 was quantified using ImageJ software. The magnification of Fig. 3a is consistent, and we supplemented the scale with 20uM. Exogenous Nesfatin-1 was intervened in BeWo cells and PHT cells for 4 h according to sections 4.2 and 4.3 of Materials and Methods. I determined the percentage of fusion rate by E-cadherin immunofluorescence staining as shown in Figure 3 a and Figure 4 a, respectively. Also, the magnification of Fig. 4a is consistent with his scale of 10 uM. Additionally, we have added instructions in section 2.6, as shown in lines 246-271 of the revised manuscript.

  • In the discussion section, in line 264, what is the maternal blood trophoblast? In lines 287-289, the clarity is lacking. For lines 356-359, there is nothing in the results section, connecting the EGFR-PLCG1-CAMK4 pathway and related genes. Overall, the discussion section requires some improvements.   

Response: Many thanks for your careful check. Further review of the literature confirms that the maternal trophoblast is the circulating trophoblast in maternal blood. Changes have been made, as described in lines 276-277 of the revised manuscript. In addition, we have added a description of the EGFR-PLCG1-CAMK4 signaling pathway in the “Results” section, as described in lines 249-274 of the revised manuscript.

Reviewer 2 Report

Comments and Suggestions for Authors

In the present work, Dang et al. try to verify that NUCB2/Nesfatin-1 contributes to trophoblast cell fusion in placenta development via the EGFR-PLCG1-CAMK4 pathway. However, some questions also should be explained.

1. Editing of English language and style is needed. Please check them throughout the manuscript. I suggest you to seek the support of a Professional English proofreading and editing service before submitting the revised version of your manuscript.

For example,

Line 18, “We found that when both primary human trophoblast”.

Line 32, “the syncytialization of trophoblast cells plays a critical role”.

Line 36, “cytotrophoblast (CTB) cells must unite”.

Line 37, “and nutritional functions that plays an”.

 Lines 56-57, “and that serum Nesfatin-1 levels were negatively associated with the presence and severity of preeclampsia (PE) [15, 16].”.

Lines 77-83, “We performed double immunofluorescence labeling of E-cadherin and NUCB2 in the rat placenta at different times, and the results showed that the fluorescence area of E-cadherin changed in the labyrinthine and connective areas from gestational day 14.5 to 19.5 in a consistent trend, with a significant decrease on gestational day 17.5 compared with that on gestational day 14.5 (P<0.001), and a significant increase on gestational day 19.5 compared with that on gestational day 17.5 (P<0.001).”. There are too long sentences.

There are ‘Figure’ and ‘Fig.’.

Line 150, “n = 3 in a b c d e f”.

Line 173, “n = 3 in a b c”.

Line 202, “Figure 5 a,b,c”.

Line 410, “Assay:”.

Line 414, “CO2”.

Line 453, “the2-ΔΔCt”.

Line 498, “Analysis”.

2. In general, scientific papers are written in the third-person manner rather than the first person. Please check them throughout the manuscript. There are so many ‘we’ and ‘our’.  

3. In general, there is always no citation in Results section.

4. In Figure 1a, images of DAPI are not fully corresponded to the images of E-cadherin at GD 17.5 and GD 19.5. Figure 1c and e, GD 17.5 and GD 19.5 is not fully corresponded to the blots. The bars for DAPI should be not same as the bars for DAPI for NUCB2 and E-cadherin.

5. In all Figure legends, ‘Data are expressed as mean ± SD and analyzed by one-way ANOVA and Tukey-Kramer multiple comparison test’ should be deleted.

6. Materials and Methods section should be after the Introduction section according to this Journal style.

7. Format of all references is not suitable for this Journal.

8. Nesfatin-1 also promotes trophoblast cell proliferation, migration, invasion through suppressing oxidative stress via activating PI3K/AKT/mTOR and AKT/GSK3β signaling pathway (Li et al., 2021).

Li T, Wei S, Fan C, Tang D, Luo D. Nesfatin-1 Promotes Proliferation, Migration and Invasion of HTR-8/SVneo Trophoblast Cells and Inhibits Oxidative Stress via Activation of PI3K/AKT/mTOR and AKT/GSK3β Pathway. Reprod Sci. 2021;28(2):550-561.

Comments on the Quality of English Language

English very difficult to understand/incomprehensible.

Author Response

Reviewers 2

In the present work, Dang et al. try to verify that NUCB2/Nesfatin-1 contributes to trophoblast cell fusion in placenta development via the EGFR-PLCG1-CAMK4 pathway. However, some questions also should be explained.

  1. Editing of English language and style is needed. Please check them throughout the manuscript. I suggest you to seek the support of a Professional English proofreading and editing service before submitting the revised version of your manuscript.

For example,

Line 18, “We found that when both primary human trophoblast”.

Line 32, “the syncytialization of trophoblast cells plays a critical role”.

Line 36, “cytotrophoblast (CTB) cells must unite”.

Line 37, “and nutritional functions that plays an”.

Lines 56-57, “and that serum Nesfatin-1 levels were negatively associated with the presence and severity of preeclampsia (PE) [15, 16].”.

Lines 77-83, “We performed double immunofluorescence labeling of E-cadherin and NUCB2 in the rat placenta at different times, and the results showed that the fluorescence area of E-cadherin changed in the labyrinthine and connective areas from gestational day 14.5 to 19.5 in a consistent trend, with a significant decrease on gestational day 17.5 compared with that on gestational day 14.5 (P<0.001), and a significant increase on gestational day 19.5 compared with that on gestational day 17.5 (P<0.001).”. There are too long sentences.                                                                                                                                                                                                                                                                                                                                                                                                                                                                                                                             

Line 150, “n = 3 in a b c d e f”.

Line 173, “n = 3 in a b c”.

Line 202, “Figure 5 a,b,c”.

Line 410, “Assay:”.

Line 414, “CO2”.

Line 453, “the2-ΔΔCt”.

Line 498, “Analysis:”. 

Response: Many thanks for this comment. We have edited the full text for English language and style.

  1. In general, scientific papers are written in the third-person manner rather than the first person. Please check them throughout the manuscript. There are so many ‘we’ and ‘our’.  

Response: Many thanks for this comment. We have changed the first person in the text to the third person, as described in the revised manuscript

  1. In general, there is always no citation in Results section.

Response: Many thanks for this comment. We have removed the citation references in the results section.

  1. In Figure 1a, images of DAPI are not fully corresponded to the images of E-cadherin at GD 17.5 and GD 19.5. Figure 1c and e, GD 17.5 and GD 19.5 is not fully corresponded to the blots. The bars for DAPI should be not same as the bars for DAPI for NUCB2 and E-cadherin.

Response: Many thanks for your careful check. We modified the markers in Fig. 1a and the markers in Figs. 1c and e. Also after checking, the scale bar in the second row corresponding to the original DAPI was changed to 1uM.

  1. In all Figure legends, ‘Data are expressed as mean ± SD and analyzed by one-way ANOVA and Tukey-Kramer multiple comparison test’ should be deleted.

Response: Many thanks for this comment. In all figure legends, "Data are expressed as mean ± SD and analyzed by one-way ANOVA and Tukey-Kramer's multiple comparison test" has been removed.

  1. Materials and Methods section should be after the Introduction section according to this Journal style.

Response: Many thanks for this comment. I wrote the manuscript following the template for this journal, in which the Recommended Materials and Methods section is placed after the Discussion section.

  1. Format of all references is not suitable for this Journal.

Response: Many thanks for this comment. We have modified the references format accordingly, as described in the revised manuscript.

  1. Nesfatin-1 also promotes trophoblast cell proliferation, migration, invasion through suppressing oxidative stress via activating PI3K/AKT/mTOR and AKT/GSK3β signaling pathway (Li et al., 2021).

Li T, Wei S, Fan C, Tang D, Luo D. Nesfatin-1 Promotes Proliferation, Migration and Invasion of HTR-8/SVneo Trophoblast Cells and Inhibits Oxidative Stress via Activation of PI3K/AKT/mTOR and AKT/GSK3β Pathway. Reprod Sci. 2021;28(2):550-561.

Response: Many thanks for this comment. I have modified the reference format and cited the literature in the abstract, as seen in the revised manuscript.

Reviewer 3 Report

Comments and Suggestions for Authors

The authors produce a high-quality manuscript with very weighty results. This manuscript represents an important advance in this field. Authors must make minor changes for final acceptance:

-The authors must include some key words from the aspect of embryology.

-The authors must improve the description of the figures, in the figures legends.

-Figure 6 must be divided into two figures.

-The panel in figure 6.A should be better described.

-The authors must justify the use of statistical tests.

-The authors must include the role of extracellular vesicles in the discussion; they must include the manuscript doi: 10.3389/fcell.2022.1060850.

-Authors must include a graphic summary.

Author Response

Reviewers 3

Comments and Suggestions for Authors

The authors produce a high-quality manuscript with very weighty results. This manuscript represents an important advance in this field. Authors must make minor changes for final acceptance:

-The authors must include some key words from the aspect of embryology. 

Response: Many thanks for this comment. We have added “embryonic development” to our keywords.

-The authors must improve the description of the figures, in the figures legends. 

Response: Many thanks for this comment. We've rewritten our figure legend, see revised manuscript for details.

-Figure 6 must be divided into two figures. 

Response: Many thanks for this comment. We have split the original Figure 6 into Figure 6 and Figure 7, as seen in the revised manuscript.

-The panel in figure 6.A should be better described. 

Response: Many thanks for this comment. We have added a description of Figure 6, as described in the revised manuscript line 246-271.

-The authors must justify the use of statistical tests. 

Response: Many thanks for your careful check. We have added instructions to the “4.10. Statistical analysis section”, as described in line 598 of the revised manuscript.

-The authors must include the role of extracellular vesicles in the discussion; they must include the manuscript doi: 10.3389/fcell.2022.1060850. 

Response: Many thanks for this comment. We have added the doi number to the reference formatting. We have added a discussion of extracellular vesicles to the discussion, as described in lines 356-362 of the revised manuscript.

-Authors must include a graphic summary. 

Response: Many thanks for this comment. We have completed the graphic summary, which can be found in the supplemental materials.

Round 2

Reviewer 2 Report

Comments and Suggestions for Authors

Thanks for author’s responses. However, editing of English language and style is still needed.

 For example, ‘P<0.05’, or ‘p<0.05’, or ‘p < 0.05’. A Editing Certificate may be needed.

Comments on the Quality of English Language

Extensive editing of English language required.

Author Response

Reviewers 2

Thanks for author’s responses. However, editing of English language and style is still needed. For example, ‘P<0.05’, or ‘p<0.05’, or ‘p < 0.05’. A Editing Certificate may be needed.

Response: Many thanks for your careful check. We have retouched the language in the text in MDPI's retouch system and double-checked the entire text again, making appropriate changes to the issues you mentioned, as detailed in the revised manuscript.
